# Ultrathin wide-angle large-area digital 3D holographic display using a non-periodic photon sieve

Jongchan Park[1,2], KyeoReh Lee [1,2] & YongKeun Park[1,2,3]

Holographic displays can provide a 3D visual experience to multiple users without requiring special glasses. By precisely tailoring light fields, holographic displays could resemble realistic 3D scenes with full motion parallax and continuous depth cues. However, available holographic displays are unable to generate such scenes given practical limitations in wavefront modulation. In fact, the limited diffraction angle and small number of pixels of current wavefront modulators derive into a 3D scene with small size and narrow viewing angle. We propose a flat-panel wavefront modulator capable of displaying large dynamic holographic images with wide viewing angle. Specifically, an ultrahigh-capacity non-periodic photon sieve, which diffracts light at wide angles, is combined with an off-the-shelf liquid crystal display panel to generate holographic images. Besides wide viewing angle and large screen size, the wavefront modulator provides multi-colour projection and a small form factor, which suggests the possible implementation of holographic displays on thin devices.

[1] Department of Physics, Korea Advanced Institute of Science and Technology, Daejeon 34141, Republic of Korea. [2] KAIST Institute for Health Science and Technology, KAIST, Daejeon 34141, Republic of Korea. [3] Tomocube, Inc, Daejeon 34051, Republic of Korea. Correspondence and requests for materials should be addressed to Y.P. (email: yk.park@kaist.ac.kr)

Holographic displays capable of generating arbitrary wavefronts can be considered as the ultimate 3D visualization technology. In fact, wavefronts, which are 2D maps of spatial phase distribution, can encode directional information of light fields, and their controlled convergence and divergence during propagation generate 3D scenes. Unlike 2D image-based stereoscopic displays, which reportedly cause headache, eyestrain, visual discomfort, and fatigue to users given vergence–accommodation conflicts[1], holographic displays are comfortable for the user and provide realistic 3D scenes with full motion parallax and continuous depth cues[2–4]. Commonly used approaches for 3D visualization are volumetric displays, which have components distributed in a volume to scatter light in every direction and thus create realistic 3D scenes[5,6]. However, the required volume for such displays makes them cumbersome and thus limits their applicability. In contrast, 3D images from holographic displays are generated using a 2D surface by exploiting the wave nature of light.

Ideal holographic displays can precisely tailor wavefronts beyond the human perception, and therefore resemble 3D scenes that are indistinguishable from real objects. However, current wavefront modulators, such as spatial light modulators (SLMs), cannot handle the large-optical modes required for generating realistic 3D scenes. Unlike real objects, whose light scatters in every direction and can thus be observed from any perspective, the viewing (i.e., twice of diffraction) angle of a 3D scene generated using the available SLMs is limited to few degrees, specifically, $\theta = 2\sin^{-1}(\lambda/2p)$, where $\lambda$ is the wavelength of light and $p$ is the pixel pitch of the display. Additional magnification lenses may increase the viewing angle with a trade-off on the scene size. In practice, the space–bandwidth product (SBP), which is a measure to determine the size and viewing angle of holographic images, is fundamentally limited by the number of addressable optical modes (i.e., pixels) when using SLMs[7].

Spatial[8–11] or temporal[12–14] multiplexing of SLMs notably increases the controllability of light fields to generate realistic 3D scenes. However, the large computational cost and transportation burden of holographic data restricts real-time operation. In addition, the resulting bulky and complex systems hinder commercialisation. On the other hand, eye-tracking[15] and head-mount[16] holographic displays restrict the viewing zones to provide improved 3D scenes, but they cannot be used for multi-user platforms. Furthermore, rewritable photopolymers[17,18] and reprogrammable metasurfaces[19] address a large number of optical modes with high-spatial density, but compromising the dynamic modulation capabilities given time-consuming refreshing processes. To date, no wavefront modulator allows displaying 3D dynamic holograms with wide viewing angle and large image size to multiple users.

Unlike previous developments, we propose a flat-panel wavefront modulator for generating 3D dynamic holographic images endowed with large-area and wide viewing angle. The key idea is the use of an ultrahigh-capacity non-periodic photon sieve consisting of randomly oriented small pinholes. By placing the photon sieve close to a transmissive liquid crystal display (LCD) panel, the diffraction angle of a light field can be considerably enhanced (Fig. 1). In addition, a one-to-one linear relation between pinholes and pixels of the LCD panel allows full independent modulation of the light field scattered from each pinhole to generate the dynamic holographic images. Furthermore, the proposed technique maintains the small form factor of the LCD panel. Likewise, dynamic colour images are rendered using a single modulator and without requiring colour filters.

## Results

**Non-periodic photon sieve.** The proposed wavefront modulator confines the active area of each LCD pixel through pinholes of the photon sieve (Fig. 2a–c). The pinholes diffract the light fields at wide angles[20,21], thus increasing the viewing angle of the holographic images. We designed the photon sieve such that its pinholes show a one-to-one correspondence with the pixels; only one pinhole is positioned on the active area of each LCD pixel, and the total number of pinholes are same as the pixel numbers in LCD panel (Fig. 2a). The probability distribution of the lateral ($x$–$y$ plane) displacement of the pinhole position from the centre position of the LCD pixel is a continuous uniform distribution. The optical field scattered from each pinhole is independently modulated by each LCD pixel and render dynamic holographic images.

The operation of the proposed system relies on the randomness of the linear optical transformation induced by the non-periodicity of the pinholes. In practice, if pinholes are periodically aligned, reconstructed holograms can only carry optical information within the Nyquist frequency defined by the pixel pitches. As a result, spatial aliasing artefacts occur; unwanted multiple cloned images are generated. In contrast, pseudo-randomly oriented pinholes suppress the duplication of the holographic images within the diffraction angle defined by the pinhole size, but at the expense of background noises (see Supplementary Figure 1 and Supplementary Note 1).

Importantly, the positions and shapes of the pinholes in the photon sieve are completely known and do not change over time, and the light transport through the LCD panel and the photon sieve occurs in a deterministic manner. Furthermore, because each pinhole is located within the area of each corresponding LCD pixel, the present method does not require complex time-consuming calibration and calculation associated with an optical transmission matrix[22,23].

We should note that our methodology is fundamentally different from previously reported photon sieve based optical holograms. The previous methods have tailored the sizes and distributions of the pinholes for generating static holograms[24,25] with given design principles. In contrast, we tailor the light field

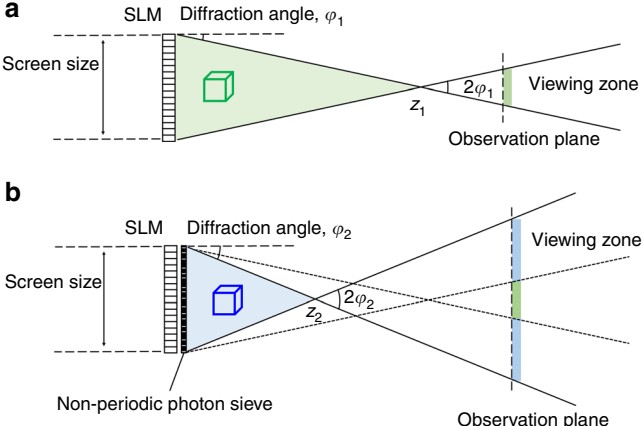

**Fig. 1** The concept of the holographic display using a non-periodic photon sieve. **a** Schematic of a lensless digital holographic display system using a single spatial light modulator (SLM). The small diffraction angle of the SLM limit the viewing angle of the holographic objects. **b** Schematics of a large-sized wide viewing angle holographic display system using a non-periodic photon sieve. The photon sieve diffracts light at a high angle, thus increases the viewing angle. To guarantee the whole holographic image can be observed without disruption within the viewing angle, the holographic objects should be positioned in the shaded region. Outside the region, the holographic objects are clipped, and the viewing angle is decreased

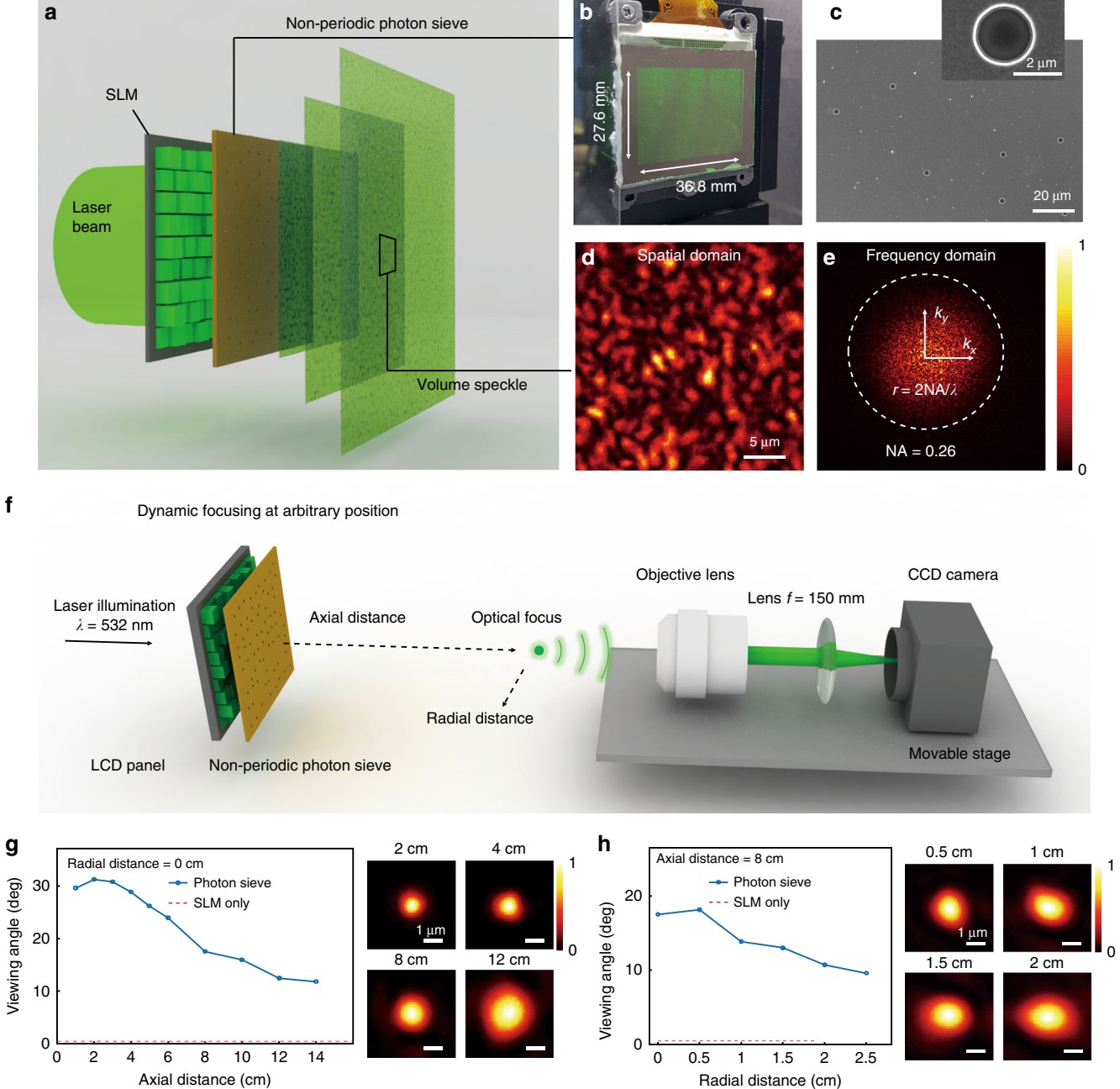

**Fig. 2** Wide-angle large-area holographic display using non-periodic photon sieve. **a** Scheme of the system. The non-periodic photon sieve increases the diffraction angles of transmitting light. A one-to-one correspondence between SLM pixels and pinholes allows independent modulation of the optical field transmitted from each pinhole. **b** Photograph of the system. The pinholes are placed close to the transmissive SLM to maintain the small form factor. **c** Micrographs from a scanning electron microscope of the non-periodic photon sieve fabricated using conventional photolithography. The diffraction angles of the transmitting light field vary according to the pinhole size. **d** Intensity profile of light transmitted through the non-periodic photon sieve. The size of the pinholes in the photon sieve is 2.2 μm. The distance between the image plane and the photon sieve is 10 mm. **e** Spatial frequency map corresponding to the intensity pattern of the image in **d**. The dashed circle indicates the numerical aperture corresponding to the viewing angle of 30 degrees. **f** Scheme of dynamic optical focusing over a wide volume. Optical focus is generated at arbitrary positions by displaying optimal patterns at the LCD panel. A 4-*f* telescopic imaging system (NA = 0.75) is mounted on a movable stage to capture images. **g**, **h** The size and shape of the focus vary according to its position because the addressable transversal wave vector depends on the displaying geometry. Effective viewing angles of the foci were examined from the size of the foci

incident on the pinholes to generate dynamic holograms while the lateral positions of the pinholes are designed in a pseudo-random manner. This pseudo-random optical transformation by the pinholes enables to manipulate holograms with a large degree of freedom.

**Experimental verification.** To verify the generation of holographic images with a wide viewing angle, we first measured a light field scattered from the photon sieve, which was fabricated using a conventional photolithography process (see Methods). The size of the pinholes in the photon sieve is 2.2 μm (Fig. 2c).

When the photon sieve is close to the LCD panel (1.8 inches diagonal, $1024 \times 768$ pixels) and illuminated by a collimated green laser beam ($\lambda = 532$ nm), a random speckle pattern is generated (Fig. 2d). As shown in Fig. 2e, the speckle pattern contains high-spatial-frequency components capable of providing wide viewing angles to the holographic images.

Next, we verified the dynamic focusing at arbitrary positions, as illustrated in Fig. 2f. An optimal pattern is displayed on the LCD panel to produce the optical focus at the desired position. We determined the phase values, which match the optical path lengths of the light fields scattered from the pinholes to the desired focal position, by simple algebraic calculations (see Methods). The resulting bias values corresponding to the phase values of the patterns were displayed on the LCD panel. In practice, the LCD panel is neither amplitude- nor phase-only modulator. The randomness of the photon sieve scrambles the optical properties of the propagating light fields and allows the conventional LCD panel to be used as an efficient wavefront modulator upon precise calibration (Supplementary Figure 2). By using a 4-$f$ telescopic imaging system with a high numerical aperture (NA = 0.75), we observed the tight optical focus as shown in Fig. 2g and Fig. 2h.

To examine an effective viewing angle of the proposed system, we measured the size of the optical focus. For a given diffraction-limited imaging system with a circular aperture, a shape of an optical focus follows an Airy pattern. A full-width at half-maximum (FWHM) value of the pattern is directly related to the NA (or viewing angle) of the imaging system as FWHM = $0.51\lambda$/NA. Along the axial direction, FWHM values of the foci at 1, 2, 3 cm from the screen were measured as 1.0 μm. The corresponding viewing angle is approximately 30° (Fig. 2g). However, when the distance between the screen and the focus increases over a certain distance ($Z_{1,2}$ in the Fig. 1), the effective viewing angle decreases inversely proportional to the distance since the supported spatial frequency range is limited by the geometry rather than diffraction angles of the pinholes. Similarly, the radial position of the focus imposes asymmetry in the addressable spatial frequency ranges for the radial directions, thus changes the shape and size of the focus[6]. Also, the non-uniform diffraction of the laser beam through the pinholes also may decrease the effective viewing angles.

It is worth noting that unlike the previous research on wavefront shaping techniques that consider disordered systems[22,26–30], we used a deterministically generated pseudo-random mask to exploit randomness without requiring a time-consuming calibration process[31]. Furthermore, the absence of relaying optics between the pinholes and the LCD panel results in minor aberrations, induced by the LCD panel and the glass substrate of the metal mask, and should be calibrated only once for optimal performance. Although the holographic images were formed in a large-optical-mode space ($>10^9$), the complexity of the linear optical transformation, i.e., the number of non-zero entries of the transmission matrix of the photon sieve, is equal to the number of pixels on the LCD panel ($\sim 10^6$). Thus, the calculation of the optimal pattern for focusing requires only $O$ ($\sim 10^6$) rather than $O(>10^9)$ computations.

**Holographic image generation**. To confirm the 3D nature of the holographic images, we generated and captured a helix hologram consisting of 75 points at different viewing angles (Fig. 3). The helix was positioned 4.2 cm behind the photon sieve, which consists of 2.2 μm sized pinholes. For this experiment, we mounted the 4-$f$ telescopic imaging system (NA = 0.16) on a movable stage to observe the motion parallax of the hologram. Figure 3b clearly shows that the

images of the hologram vary according to the observation angle, and the experimental results are consistent with those from simulations. Although the actual viewing angle was measured as 30°, the holographic images can be observed at higher angles because our imaging system has a non-zero numerical aperture. It is worth noting that we achieved a wide viewing angle of about 30° using a single flat-panel wavefront modulator without additional optics. By using the smaller sized pinholes, the viewing angle can be further enhanced (Supplementary Figure 3; Supplementary Figure 4).

To demonstrate the projection of holographic images on a large volume, we increased the size of the helix to the centimetre scale (Fig. 3c). The diameter of the helix varies along the axial direction, and the helix has maximum diameter and length of 0.5 cm and 5 cm, respectively (Fig. 3d). We captured the 2D projections of the hologram by placing a camera without lenses at each depth plane and observed the corresponding images. In Fig. 3d, the blurred images of out-of-plane foci are also shown. In practice, although the large screen size and the wide diffraction angles were confirmed, the actual viewing angles of holographic images are restricted by its displaying geometries. The holographic images must be lie between the observers' eyes and the scattering surface (the photon sieve) since the light is emitted from the displaying screen rather than the virtual objects[32].

A contrast factor, defined as the ratio between the intensity of the holographic image and the average value of the background noise, is directly proportional to the number of controlled optical modes in our system. By using all the pixels in the XGA resolution ($1024 \times 768$ pixels) of the LCD panel, we achieved a factor of approximately $2.0 \times 10^5$ (Supplementary Figure 5), thus notably outperforming previous works on wavefront shaping through disordered systems[33,34]. Although this value, to the best of our knowledge, is the highest ever reported, still insufficient to generate complex holographic images. When a large number of optical modes (or optical foci) are addressed simultaneously, the signal is redistributed into the optical modes. In that case, the signal level decreases in proportion to the number of the optical modes while the background noise remains the same. In order to maintain the high visibility of the holographic images, either the signal must be sparse or to utilise a large number of controlled optical modes (pixel numbers).

In Table 1, the performance of the LCD panel integrated with the photon sieve for displaying holographic images in terms of the screen size and diffraction angle is compared with commercially available SLMs and our previous research on a 3D display[26]. The values of the proposed system in Table 1 are achieved using a non-periodic photon sieve of 2.2 μm sized pinholes. In our demonstration, the product value of the screen size and diffraction angle was enhanced over a factor of ~1200 compared to the use of only a wavefront modulator with XGA resolution while the holographic image generated from our display attain the same SBP (or complexity) value (see Discussion).

We also verified the generation of dynamic colour holograms using a single wavefront modulator by presenting a rotating cube with three colours, as shown in Fig. 4. Red ($\lambda = 639$ nm), green ($\lambda = 532$ nm), and blue ($\lambda = 473$ nm) laser beams illuminated the LCD panel simultaneously, and the optimal patterns were displayed to generate coloured foci at different spatial positions. The optical transformations of the photon sieve are spectrally uncorrelated for the given wavelengths. Therefore, the colour images were projected simultaneously using the modulator resembling space-division multiplexing such that no colour filter was required[35] (see Methods).

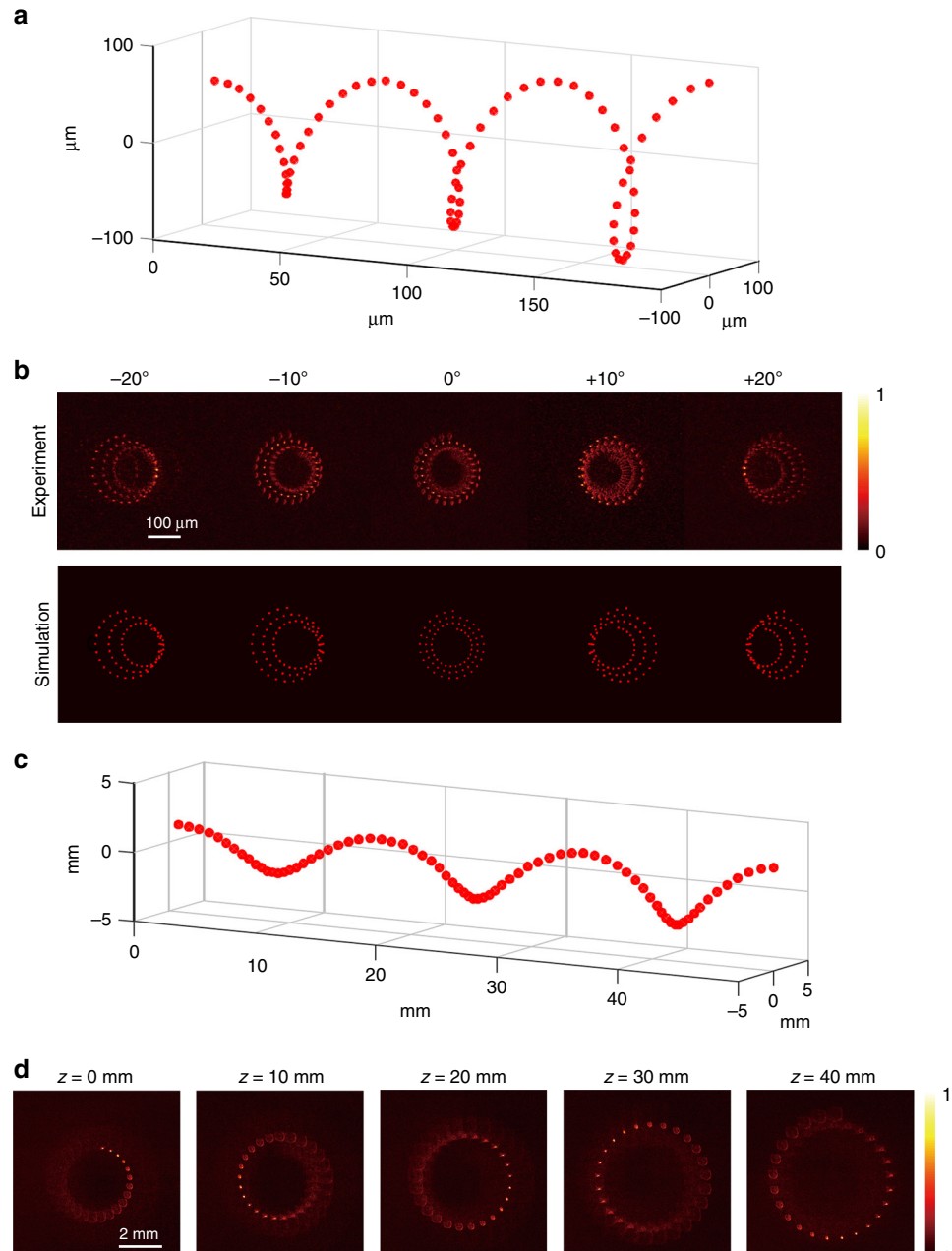

**Fig. 3** Holographic images of 3D helixes. To demonstrate the motion parallax of holographic images, a helix consisting of 75 foci was generated and placed within a depth of focus of the proposed imaging system. **a** Geometry of the holographic helix. **b** 2D intensity images captured at different observation angles using the 4-*f* telescopic imaging system with a numerical aperture of 0.16 and corresponding simulation results. **c** Holographic image for projection in a large volume. The diameter of the helix varies from 0.25 cm to 0.5 cm with a length of 5 cm. **d** 2D intensity maps of the helical trajectory captured at each depth without lenses mounted on the camera

**Varying pinhole size and number of pinholes per LCD pixel**. Because the active area of the LCD pixel is confined by the size of the pinholes, the photon sieves are suffered from low light efficiency. In our demonstration, the light transmittance through the photon sieve, which consists of 2.2 μm sized pinholes, was measured as 0.16%. To mitigate the issue, we demonstrated the proof of concept experiments with the photon sieves consisting of either large-sized pinholes or larger number of pinholes.

As shown in Fig. 5a, the transmittance of the proposed method is readily adjustable upon demand by compromising the viewing angles. We should note that although the increased transmittance was achieved in trade relations between diffraction angles. Still, the photon sieves notably increase the viewing angle of the

transmissive LCD, which was originally 0.8 degree. Compared to the use of reflective SLMs, it is more challenging to narrow down the pixel pitch of the transmissive SLMs because of the embedded electrodes and required optical pathlengths of rotating birefringent molecules. Until now these technical hurdles obstruct the realization of flat-panel holographic displays with large viewing angles. The larger the number of pinholes, the higher the transmittance (Fig. 5b). However, the calculation burden also increases linearly with the number of pinholes. Nevertheless, we successfully demonstrated the generation of holographic images, a tetrahedron consisting of 60 foci, with a total of 7.8 million pinholes (Fig. 5d). The holographic image converges and diverges rapidly. The shape highly depends on the observing planes, which

**Table 1 Characteristics of the holographic displays**

| System | Pixel pitch | Pixels (modes) | Screen size | Viewing angle | Contrast factor |
|---|---|---|---|---|---|
| Hamamatsu, phase-only SLM | 20 μm | 800 × 600 | 1.2 cm × 1.6 cm | 1.5° | – |
| Texas Instruments, DMD | 5.4 μm | 1920 × 1080 | 1.04 cm × 0.58 cm | 5.6° | – |
| Holoeye, 10 Megapixel SLM | 3.74 μm | 4160 × 2464 | 1.60 cm × 0.94 cm | 8.2° | – |
| Holoeye, transmissive SLM | 36 μm | 1024 × 768 | 3.68 cm × 2.76 cm | 0.8° | – |
| System by Yu et al.[26] | ~0.5 μm[a] | 128 × 128 | 4.2 cm in diameter | ~60° | ~1671 |
| Proposed system | ~2.2 μm[b] | 1024 × 768 | 3.68 cm × 2.76 cm | ~30° | ~200,000 |

Characteristics of the holographic display with the non-periodic photon sieve are compared with that of the commercially available SLMs, and the previously proposed holographic display. A graphical illustration is provided in Supplementary Figure 6.
[a]Effective pixel pitch corresponding to the diffraction angle
[b]Pinhole size

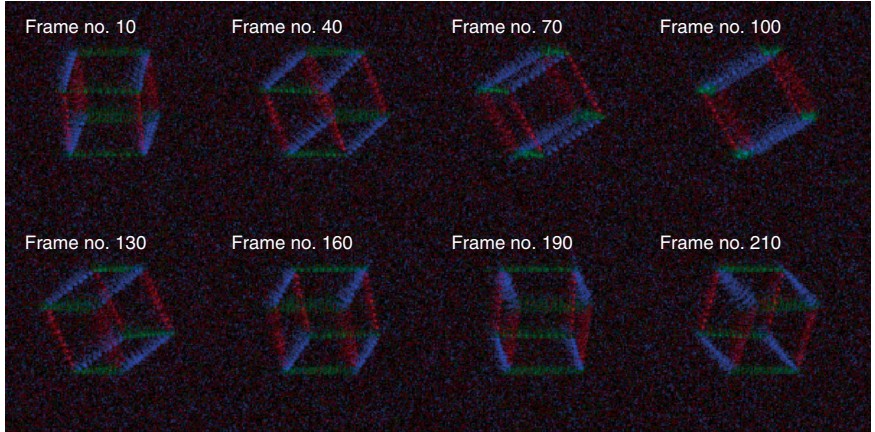

**Fig. 4** Dynamic colour holograms of a rotating cube. Frames of a 3D holographic rotating cube with red, green, and blue colours. See Supplementary Movie 1 for the full set of frames

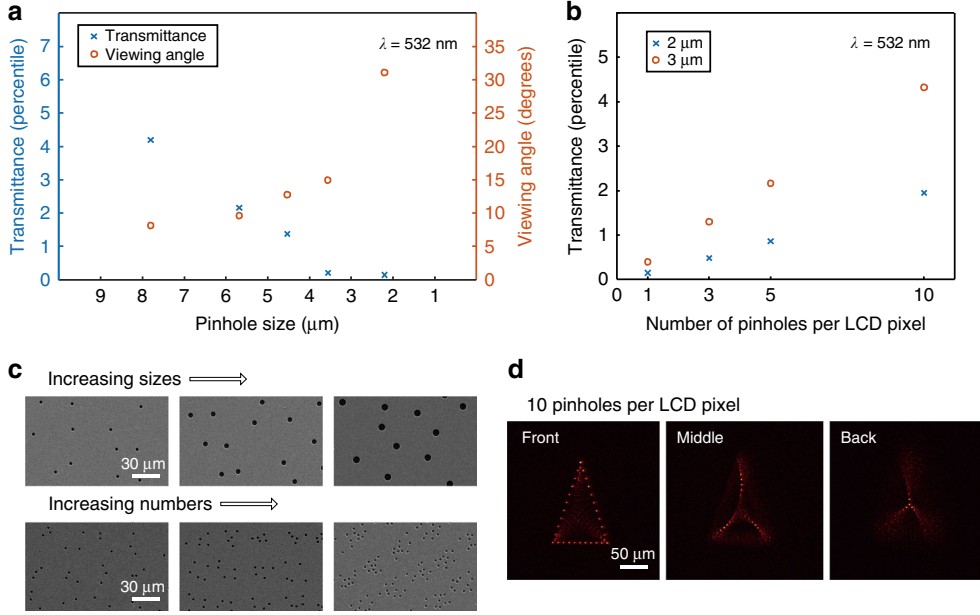

**Fig. 5** Varying sizes and numbers of pinholes. Total light transmittance of the photon sieve versus size (**a**) and numbers (**b**) of the pinholes. The number of pinholes is equal to the number of the LCD pixels in **a**. **c** Scanning electron microscope images of the photon sieves. **d** Holographic tetrahedron projected by using the photon sieve having 10 pinholes per LCD pixel

reflect the nature of the large diffraction angle of the holographic display.

## Discussion

Although the demonstrated holographic imaging with a wide viewing angle in a large-area surpasses the capability of the conventional wavefront modulators, these results do not violate the Nyquist–Shannon sampling theorem. Instead, the optical signal (i.e., holographic image) containing $N$ optical modes, which correspond to the number of pixels on the LCD panel, is pseudo-randomly redistributed into a larger optical mode space in a deterministic way at the expense of background noises. Given the one-to-one linear relation between pixels and pinholes, the rank, which is the number of linearly independent columns of the optical transmission matrix of a photon sieve, is $N$. Therefore, the 3D holographic images generated using the photon sieve can be fully decomposed into $N$ orthogonal optical modes (samples) regardless of the wide viewing angle and large displaying area.

We note that a product of a total size of the screen and the diffraction angle in the holographic display is not necessarily same as the SBP (or information capacity) of the optical signal generated from it. The SBP is considered as one of the most important metrics for evaluating the quality of holographic displays and previously often regarded as the product value of the image sizes and viewing angles. However, the SBP is defined as the product of the spatial size and frequency range within which the signal is non-zero. When an optical field is scattered by the pinholes, the non-zero area of the optical field is confined to small by the pinholes. However, the total screen size remains same. At the same time, the small size of the pinholes broaden the spatial frequency ranges. Therefore, the resultant holographic image can exhibit a large-size with a wide viewing angle, but possess a small SBP value at the same time.

Our demonstrations indicate that if a holographic image is sparse, it can be displayed with high fidelity by a pseudo-random optical projection even if the image is projected on a large-optical-mode space. In practice, the random projection methods in imaging systems have actively been researched in the field of the compressed sensing[36,37] because sparse signals can be dealt with a small number of sampling measurements with prior knowledge or assumptions. However, little has been reported for displaying purpose since the inherent noises in the displaying signals induced by the randomness cannot be removed without additional filtering or processing. Nevertheless, we believe that this random projection approach suggests a very promising way in displaying sparse signals with a limited number of controllable degrees of freedom, especially for holographic display applications where the amount of holographic information usually exceeds the modulation capabilities of the current electronic devices.

Although promising, the proposed system shows practical limitations such as low power efficiency and incomplete non-periodicity. In fact, the photon sieve blocks most of the light field, thus requiring the high-power laser to keep the visibility of holographic images. Likewise, the marginal periodicity of the pinholes caused by the limited fill factor of available LCD panels might generate cloned images of relatively low intensity. These limitations could be mitigated by using other advanced disorder-manipulated systems such as micro-sized gratings with varying pitch and orientation, micro-lens arrays with spatially varying shapes, and combinations of them with photon sieves.

In conclusion, we present a proof-of-concept for a flat-panel wavefront modulator to generate large-size wide-viewing-angle dynamic 3D holographic images. An easily manufactured non-periodic photon sieve is combined with a commercial LCD panel

to generate the images with arbitrary viewing angle and image size while maintaining the LCD small form factor. Our technique can be easily integrated into the current LCD production process and is a promising approach towards thin holographic displays. The proposed programmable randomly distributed system is not only limited to holographic displays but also enables versatile light field control.

## Methods

**Experimental system**. We used a 1.8-inch transmissive LCD panel (LCX017AL, Sony Co., Tokyo, Japan) with XGA resolution of 1024 × 768 pixels to modulate the incident wavefront of a coherent laser beam (Fig. 2a) at a maximum modulation speed of 60 Hz, which was limited by the LCD panel. By using a double-slit interferometer, the optical response of the LCD panel was precisely measured according to the applied bias (Supplementary Figure 3). We selected the polariser and analyser orientations which maximize the phase delay induced by the LCD panel (the maximum achievable phase delay using the LCD panel at $\lambda = 532$ nm is $1.57 \pi$).

We mounted the LCD panel on a custom-made holder and precisely aligned the photon sieve to be in physical contact with the panel. Four square-shaped markers were patterned at the four corners of the photon sieve and were utilised for initial alignment. Then, we placed a charge-coupled device (CCD) camera (Lt365R, Lumenera Co., Ottawa, Canada) 20 cm behind the photon sieve and updated a pattern, which focuses the beam on the CCD, on the LCD panel. We precisely adjusted the translational positions of the photon sieve that maximise the intensity value detected at the CCD.

To compensate the aberration induced by the LCD panel itself and the glass substrate of the photon sieve, we applied a correction pattern to the panel. The correction pattern was formed by successively adding the Zernike polynomials of up to order of 21st ($\mathbf{Z}_j$, $j = 2 \ldots 21$). The Zernike coefficients of the pattern were obtained by maximising the intensity of the focus through an iteration-based feedback loop.

In order to determine the phase map that generates foci at desired focal positions, we algebraically calculate the phase values which match the optical path length of the light fields scattered from the pinholes. The optical path lengths of light fields scattered from the pinholes to the desired focal positions are given as:

$$s_{l,m} = \sqrt{(x_l - x_m)^2 + (y_l - y_m)^2 + z_m^2},$$

where $(x_l, y_l, z_l)$ and $(x_m, y_m, Z_m)$ are the position of the $l$th pinholes and the $m$th focal positions, respectively. The corresponding phase values to be applied to the pixels of the LCD panel were found by taking the argument of linear summation of the optical field values.

$$\varphi_l = \arg\left(\sum_m a_m \exp\left(j \frac{2\pi}{\lambda} s_{l,m}\right)\right),$$

where $a_m$ is the relative amplitude of the focus.

We used a green laser ($\lambda = 532$ nm; LSS-0532, Laserglow Technologies, Toronto, Canada) for all the experiments, except for that of the coloured holographic images (Fig. 4), which also included red ($\lambda = 639$ nm; MLL-FN-639, CNI Optoelectronics Tech. Co., Ltd., Changchun, China) and blue ($\lambda = 473$ nm; MSL-FN-473, CNI Optoelectronics Tech. Co., Ltd., Changchun, China) lasers. The laser beams were combined by using dichroic mirrors (DMLP505, DMLP550, Thorlabs Inc., NJ, USA) to simultaneously illuminate the LCD panel. Moreover, the optical response of the LCD panel at each wavelength was precisely calibrated. To render the colour images using a single LCD panel, we adopted space-division multiplexing. Specifically, the pixels on the LCD panel were randomly and evenly assigned to three groups, which correspond to the red, green, and blue laser beams. Then, the wavefronts from each group were independently modulated to generate the three-colour holographic images.

**Fabrication of non-periodic photon sieve**. To fabricate the photon sieve, a Ti thin film of 300 nm was deposited on a fused silica substrate by using an electron-beam evaporator (IPE801, INFOVION Inc., Seoul, Republic of Korea). The photon sieve with 2.2 μm pinholes (Fig. 2c) was fabricated using a conventional i-line photo-lithography process (NSR-2205i11D, Nikon Co., Tokyo, Japan). Smaller 400 nm pinholes (Fig. 2c) were fabricated using a direct electron-beam writing process (JBL-9300Fs, JEOL Ltd., Tokyo, Japan).

**Image acquisition**. We employed a 4-$f$ telescopic imaging system to capture the images shown in the figures. An objective lens (NA = 0.75; UPlan FLN 40×, Olympus Co., Tokyo, Japan) and a tube lens ($f = 100$ mm) projected the optical field to a charge-coupled device (Lt365R, Lumenera Co., Ottawa, Canada) to acquire the high-resolution images in Fig. 3. For Fig. 4 and Fig. 5, we used an objective lens with a relatively low numerical aperture (NA = 0.16; UPLSAPO 4×, Olympus Co., Tokyo, Japan) to achieve a large depth of focus. In addition, we mounted the complete 4-$f$ imaging system on a movable stage to capture images from different observation angles and positions.

## Code availability

Codes used for this work are available from the corresponding author upon reasonable request.

## Data availability

The data that support the plots within this paper and other findings of this study are available from the corresponding author upon reasonable request.

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

## Acknowledgements

The authors acknowledge Da-Jin Kim in Korea Advanced Institute of Science and Technology for help in photon sieve fabrication process. This work was supported by National Research Foundation of Korea (2017M3C1A3013923, 2015R1A3A2066550, 2018K000396, 2018R1A6A3A01011043), and Samsung Electronics.

## Author contributions

J.P., K.L., and Y.P. conceived the initial idea. J.P. performed the experiments and analysed the data. K.L. contributed analytic tools. All authors wrote and revised the manuscript. Correspondence and requests for materials should be addressed to Y.P.

## Additional information

**Competing interests:** The authors declare no competing interests.

