## [Peer Review File · Nature Communications]

Editorial Note: This manuscript has been previously reviewed at another journal that is not operating a transparent peer review scheme. This document only contains reviewer comments and rebuttal letters for versions considered at Nature Communications . Mentions of prior referee reports have been redacted.

Reviewers' comments:

Reviewer #1 (Remarks to the Author):

[redacted]. I thank the authors for making effort to take on board my earlier comments: as far as I'm concerned the manuscript is now much clearer, with everything explained in reasonable detail.

As before, I find the concept of using a photon sieve to be a novel, simple and elegant approach to increasing the beam shaping capabilities of phase only spatial light modulators. The main drawbacks of the presented approach are a significant reduction in efficiency (reduced to 0.16% of incident laser power), and the fact that only sparse light fields can be created with high contrast. However, these drawbacks are now detailed in the present submission, and are also present in earlier work using random scattering media in place of a photon sieve. Yet the photon sieve approach also has the advantage of much simpler calibration, SLM phase calculation, and scatterer design than previous work, and of enabling implementation of a stable flat screen device. The demonstration of high numerical aperture 3 colour holographic light fields is an impressive result. However, I do note that because of the low efficiency I find mention of the prospect of applications to mobile phones highly unlikely in the present form, so I suggest removing this from the manuscript.

In summary, I suggest publication of the manuscript in Nature Communications.

Reviewer #2 (Remarks to the Author):

The authors have put in efforts to address the questions raised and made a lot of revisions to the manuscript. Now it is much clearer than before. It is an innovative work and had solved one of the critical issues in holographic display on the small viewing angle by putting a pinhole in front of each pixel of a LCD panel with a random arrangement. While solving one problem, however, the proposed solution creates another probably more serious issue: the extremely low efficiency of only 0.16%. This big drawback voids the significance of the work as a meaningful solution and the author's claim of possible implementation on mobile devices. Considering this, I cannot recommend this manuscript for publication in Nature Communications.

There are three other points for the authors to clarify:

- In Figure 2h, it is "Axial distance = 8cm "or "Radial distance = 8cm"? If it is later, is the 8cm correct?
- Since the contrast of the image is directly proportional to the number of controlled optical modes and the number of pinholes is equal to the number of pixels in the LCD, how come the image contrast could be improved dramatically from the LCD generated holograph with the same pixel numbers?
- For the Holoeye 10 Megapixel SLM, the pixel pitch is only 3.74 μ m and the pixel size could be even smaller than the pinhole size of 2.2 μ m, could the author double confirm and explain why its

diffraction angle is only 8.2 degree while the proposed system can reach 30 degree?

Reviewer #3 (Remarks to the Author):

The authors have made a significant revision of the original manuscript and addressed many of the points raised by the reviewers with a reasonable level of details.

However, it remains not clear what is the fundamental progress made by this paper:

1. It is not clear how why the authors claim their “dynamic” approach is fundamentally different from the previous photon sieve approach for “static” images. The randomised or non-periodic photon sieves just act as a transfer matrix. It is not necessary for the photon sieves using for “static” images to be optimised every time in order to be used for dynamic images.

2. In addition, despite the images were constructed using holograms, however, this approach may well limit the ability of image reconstruction in space with adequate depth to be called as a 3D image.

Overall, this is a nice piece of work to demonstrate certain abilities of using a non-periodic photon sieve to provide a wide angle of view with a relatively large screen size. However, this is intrinsically at the cost of image quality and optical efficiency. These limitations are not clearly explained in the manuscript to provide a balanced picture. As it is, it will be difficult in any practical use.

Reviewer #1 (Remarks to the Author):

[redacted]. I thank the authors for making effort to take on board my earlier comments: as far as I'm concerned the manuscript is now much clearer, with everything explained in reasonable detail.

We thank the reviewer for the valuable comments and suggestions on our previous submission, which are helpful to strengthen our manuscript.

As before, I find the concept of using a photon sieve to be a novel, simple and elegant approach to increasing the beam shaping capabilities of phase only spatial light modulators. The main drawbacks of the presented approach are a significant reduction in efficiency (reduced to 0.16% of incident laser power), and the fact that only sparse light fields can be created with high contrast. However, these drawbacks are now detailed in the present submission, and are also present in earlier work using random scattering media in place of a photon sieve. Yet the photon sieve approach also has the advantage of much simpler calibration, SLM phase calculation, and scatterer design than previous work, and of enabling implementation of a stable flat screen device. The demonstration of high numerical aperture 3 colour holographic light fields is an impressive result. However, I do note that because of the low efficiency I find mention of the prospect of applications to mobile phones highly unlikely in the present form, so I suggest removing this from the manuscript.

In summary, I suggest publication of the manuscript in Nature Communications.

We appreciate the clear summary of our work. As the reviewer noted, we removed the mobile implementation prospect in the revised manuscript. Besides, we put effort to show that our methodology is readily scalable and provide a strategy on how the light transmittance could be improved by compromising the viewing angle and calculation burden.

Reviewer #2 (Remarks to the Author):

The authors have put in efforts to address the questions raised and made a lot of revisions to the manuscript. Now it is much clearer than before. It is an innovative work and had solved one of the critical issues in holographic display on the small viewing angle by putting a pinhole in front of each pixel of a LCD panel with a random arrangement. While solving one problem, however, the proposed solution creates another probably more serious issue: the extremely low efficiency of only 0.16%. This big drawback voids the significance of the work as a meaningful solution and the author's claim of possible implementation on mobile devices. Considering this, I cannot recommend this manuscript for publication in Nature Communications.

We thank the reviewer for the clear summary and critical comments. As shown in Figure 5 and the related text in the revised manuscript, we showed that our methodology is readily scalable and provide a strategy on how the light transmittance could be improved by compromising the viewing angle and calculation burden.

By changing the design parameters of the pinholes, the transmittance was increased, as a cost of viewing angle. Still, the viewing angle is increased by more than ten folds (both in the x- and y- directions) compared to the use of the LCD panel only. In practice, it is technically challenging to narrow down the pixel pitch of the transmissive SLMs, which must embed electrodes and possess certain thickness to retain optical path length for rotating birefringent molecules. We believe that this alternative approach would be a solution for the realisation of flat-panel large viewing angle holographic display.

Alternatively, the transmittance can be enhanced by increasing the number of pinholes per LCD pixel. In the new experiment, we have projected the holographic image using a photon sieve having ten pinholes per LCD pixel (the figures below). In this case, the calculation costs linearly increase with the number of pinholes. We successfully demonstrated the generation of holographic images using 7.8 million pinholes. We note that the image contrast is still constrained by the number of LCD pixels.

We hope that these additional experiments and modifications have strengthened the manuscript and now suitable for publication.

Figure 5 | Varying sizes and numbers of pinholes. Total light transmittance of the photon sieve versus size (a) and numbers (b) of the pinholes. The number of the pinholes are equal to the LCD pixels in a. c, Scanning electron microscope images of the photon sieves. d, Holographic tetrahedron projected by using the photon sieve having 10 pinholes per LCD pixel.

Varying pinhole size and number of pinholes per LCD pixel

Because the active area of a LCD pixel is confined by the size of the pinholes, the photon sieves are suffered from low light efficiency. In our demonstration, the light transmittance through the photon-sieve, which consists of 2.2-μm-sized pinholes, was measured as 0.16%. To mitigate the issue, we demonstrated the proof of concept experiments with the photon sieves consisting of either large-sized pinholes or larger number of pinholes.

As shown in Fig. 5a, the transmittance of the proposed method is readily adjustable upon demand by compromising the viewing angles. We should note that although the increased transmittance was achieved in trade relations between diffraction angles. Still, the photon sieves notably increases the viewing angle of the transmissive LCD, which was originally 0.8 degree.

Compared to the use of reflective SLMs, it is more challenging to narrow down the pixel pitch of the transmissive SLMs because of the embedded electrodes and required optical pathlengths of rotating birefringent molecules. Until now these technical hurdles obstruct the realization of flat-panel holographic displays with large viewing angles. The larger the number of pinholes, the higher the transmittance (Fig. 5b). However, the calculation burden also increases linearly with the number of pinholes. Nevertheless, we successfully demonstrated the generation of holographic images, a tetrahedron consisting of 60 foci, with a total of 7.8 million pinholes (Fig. 5d). The holographic image converges and diverges rapidly. The shape highly depends on the observing planes, which reflect the nature of the large diffraction angle of the holographic display.

There are three other points for the authors to clarify:

- In Figure 2h, it is “Axial distance = 8cm “or “Radial distance = 8cm”? If it is later, is the 8cm correct?

We appreciate the reviewer for pointing out the mistake. The x-axis in Figure 2h stands for the ‘Radial distance,’ and the inserted word stands for the ‘Axial distance = 8 cm’. The words in the figure have corrected in the revised manuscript.

- Since the contrast of the image is directly proportional to the number of controlled optical modes and the number of pinholes is equal to the number of pixels in the LCD, how come the image contrast could be improved dramatically from the LCD generated holograph with the same pixel numbers?

We believe that the reviewer concerns the information capacity of the holographic display. In practice, the ‘contrast’ or the ‘information capacity’ of the holographic images were not increased in our demonstration. The constructed holographic images can be fully decomposed into the same number of optical modes of the LCD pixels.

The main argument of our work is that the information capacity of the holographic display system is not necessarily same as the product of viewing angle and the total image size, but is the same as the complexity. Thus, the pseudo-random optical transformation of the non-periodic pinholes array enables us to generate large-sized and wide-viewing-angle holographic images with a given limited number of optical modes. However, as a drawback, our method is only applicable to the ‘sparse holograms.’

The only way to increase the information capacity of the hologram is to increase the number of the independently controllable optical modes (= number of the LCD pixels).

- For the Holoeye 10 Megapixel SLM, the pixel pitch is only 3.74 μ m and the pixel size could be even smaller than the pinhole size of 2.2 μ m, could the author double confirm and explain why its diffraction angle is only 8.2 degree while the proposed system can reach 30 degree?

The viewing angle of the conventional holographic display is given as, $\theta = 2 \sin^{-1}(\lambda / 2p) \approx 2(\lambda / 2p) \approx 8.2^\circ$ where λ and p are wavelength and pixel pitch, respectively [1]. This angle is corresponding to the Nyquist frequency limit, which is given by the pixel pitch of a SLM.

In practice, a SLM diffract light at higher angles up to $\theta \approx 2(\lambda / p) \approx 16.3^\circ$ (single slit diffraction). Even high-order diffraction beams appear in many cases. In other words, the light diffracts over 8.2 degrees is observable. However, if the hologram carries the spatial frequency beyond the Nyquist frequency limit, aliasing artefacts occur. Thus, the information beyond the angles of 8.2 degrees cannot be displayed without disruption of the image within the angle of 8.2 degrees.

In contrast, as shown in the Supplementary Information, our demonstration suppress the aliasing artefact by using the pseudo-random optical transformation. Thus the whole diffraction angle of the pinholes can be utilised. The theoretically achievable diffraction angle is given as $\theta \approx 2.44(\lambda / D) \approx 34.3^\circ$ (pinhole diffraction), where D is the diameter of the pinholes. The experimentally achieved viewing angle was up to 30 degrees (Figures 2).

In summary, we only have considered the zeroth-order diffraction of the holographic display for comparison. To avoid the confusion, we have modified the term ‘Diffraction angle’ to the ‘Viewing angle’ in Table 1 of the revised manuscript.

[1] Onural, Levent, Fahri Yaras, and Hoonjong Kang. "Digital holographic three-dimensional video displays." Proceedings of the IEEE 99.4 (2011): 576-589.

Reviewer #3 (Remarks to the Author):

The authors have made a significant revision of the original manuscript and addressed many of the points raised by the reviewers with a reasonable level of details.

However, it remains not clear what is the fundamental progress made by this paper:

We thank the reviewer for the efforts in reviewing our manuscript. We hope our point-by-point responses remove ambiguity.

1. It is not clear how why the authors claim their “dynamic” approach is fundamentally different from the previous photon sieve approach for “static” images. The randomised or non-periodic photon sieves just act as a transfer matrix. It is not necessary for the photon sieves using for “static” images to be optimised every time in order to be used for dynamic images.

First, we would like to distinguish its design principles. In our demonstrations, the lateral positions of the pinholes are designed in a pseudo-random manner that the lateral displacement of the pinhole position from the centre position of a LCD pixel is a continuous uniform distribution. The pseudo-random optical transformation induced by the non-periodicity of the pinholes completely scrambles the optical field transmitted through the pinholes and enables generating holograms with a large degree of freedom.

Most of the previous methods have tailored the sizes and distributions of the pinholes for generating static holograms with given design principles. As the pinholes are designed for particular static holograms, they attain fewer modulation capabilities.

The peak-to-noise ratios of our holograms are purely dependents on the number of optical modes addressed. However, even if combined with the wavefront shaping techniques, the performance of the holograms from the previous photon sieve methods are highly dependent on the design of the photon sieve and holograms of interest.

Finally, we would like to emphasize the technical difference between the ‘static holograms and ‘dynamic holograms.’ In practice, various metasurfaces have demonstrated to generate realistic static holograms and several holographic optical elements have already been commercialized. In contrast, none of the dynamic holographic display has achieved to meet the customer needs mainly due to its limited modulation capabilities (limited number of optical modes). Large viewing angles of the static holograms are not surprising at all, but it is surprising for the dynamic holograms, especially for a single flat wavefront modulator.

2. In addition, despite the images were constructed using holograms, however, this approach may well limit the ability of image reconstruction in space with adequate depth to be called as a 3D image.

In our previous demonstrations, we generated dynamic holographic images of centimetre scale (Figure 3d) which were reconstructed in the 3D space.

To address the reviewers concern, we also demonstrated the generation of a holographic image with larger depth. As shown in the figures below, the three ‘numbers’ were projected in a 3D space simultaneously. The ‘numbers’ were consisting of a total of 147 foci. The distances from the screen of the ‘numbers’ were set to 30 cm, 50 cm and 100 cm. The total sizes of the ‘numbers’ were set to be within the size of the image sensor (LT365R, Luemenra Co. 2/3” optical format) where the images were captured by the CCD camera without placing a lens.

In practice, the resolution (size) of each the foci was micrometre scale. However, for better visibility, the images were captured in defocused planes; the axial distances of the CCD planes were slightly displaced from the planes where the ‘numbers’ were projected. Although we successfully demonstrated the holographic image generation with larger depth planes, the image can be observed only when it lies between the observers’ eyes and the screen (which is the same for all the holographic display technologies).

Overall, this is a nice piece of work to demonstrate certain abilities of using a non-periodic photon sieve to provide a wide angle of view with a relatively large screen size. However, this is intrinsically at the cost of image quality and optical efficiency. These limitations are not clearly explained in the manuscript to provide a balanced picture. As it is, it will be difficult in any practical use.

We thank the reviewer for the critical comments. We agree with the reviewers that the present method possesses drawbacks associated with image quality and light efficiency. However, we strongly believe that our method (the pseudo-random projection approaches) would find practical applications as it solves one of the most critical issues in holographic displays—small viewing angles and image sizes. We believe that drawbacks have addressed in the revised manuscript as shown below. The direct statements about the drawbacks are given in red.

(the 2nd paragraph of 'Non-periodic photon sieve' section in the 'Results')

The operation of the proposed system relies on the randomness of the linear optical transformation induced by the non-periodicity of the pinholes. In practice, if pinholes are periodically aligned, reconstructed holograms can only carry optical information within the Nyquist frequency defined by the pixel pitches. As a result, spatial aliasing artefacts occur; unwanted multiple cloned images are generated. In contrast, pseudo-randomly oriented pinholes suppress the duplication of the holographic images within the diffraction angle defined by the pinhole size, but at the expense of background noises (see Supplementary Information).

(the 3rd paragraph of 'Holographic image generation' section in the 'Results')

A contrast factor, defined as the ratio between the intensity of the holographic image and the average value of the background noise, is directly proportional to the number of controlled optical modes in our system. By using all the pixels in the XGA resolution (1024×768 pixels) of the LCD panel, we achieved a factor of approximately 2.0×10^5 (see Supplementary Information), thus notably outperforming previous works on wavefront shaping through disordered systems. Although this value, to the best of our knowledge, is the highest ever reported, still insufficient to generate complex holographic images. When a large number of optical modes (or optical foci) are addressed simultaneously, the signal is redistributed into the optical modes. In that case, the signal level decreases in proportion to the number of the optical modes while the background noise remains the same. In order to maintain the high visibility of the holographic images, either the signal must be sparse or to utilise a large number of controlled optical modes (pixel numbers).

(the 1st paragraph of the 'Discussion')

Although promising, the proposed system shows practical limitations such as low power efficiency and incomplete non-periodicity. In fact, the photon sieve blocks most of the light field, thus requiring the high-power laser to keep the visibility of holographic images. In our demonstration, the light transmittance through the photon-sieve, which consists of $2.2 \mu\text{m}$ sized pinholes, was measured as 0.16%.

REVIEWERS' COMMENTS:

Reviewer #2 (Remarks to the Author):

The authors have properly addressed all questions raised and proposed a potential method in improving the efficiency of the system with experimental demonstration showed in Figure 5. The work is a good step forward towards the dynamic holographic display with much increased viewing angle and image sizes. I recommend accepting it for publication in Nature Comm.

Reviewer #2 (Remarks to the Author):

The authors have properly addressed all questions raised and proposed a potential method in improving the efficiency of the system with experimental demonstration showed in Figure 5. The work is a good step forward towards the dynamic holographic display with much increased viewing angle and image sizes. I recommend accepting it for publication in Nature Comm.

We appreciate the reviewer's comment.